# A Sense of Belonging and Help Seeking: Examining Factors Related to the Mental Health of High School Students with High Autistic Traits without Diagnosis

**DOI:** 10.3390/children10121927

**Published:** 2023-12-14

**Authors:** Tomoko Omiya, Naoko Kumada Deguchi, Takashi Asakura

**Affiliations:** 1Public Health Nursing, Faculty of Medicine, University of Tsukuba, Tsukuba 305-8575, Japan; 2Faculty of Education, Shizuoka University, Shizuoka 422-8017, Japan; naokok-tky@umin.ac.jp; 3Laboratory of Health and Social Behavior, Tokyo Gakugei University, Koganei 184-8501, Japan; asakurat@u-gakugei.ac.jp

**Keywords:** adolescence, autism spectrum disorder, high autistic traits, mental health, help-seeking style, high-school students, school belonging

## Abstract

Certain individuals are clinically undiagnosed for Autism Spectrum Disorder (ASD) but exhibit strong ASD characteristics. This study examined the differences between a control group and a “high autistic traits” group involving individuals who scored 9 or higher on the Autism-Spectrum Quotient-16 Japanese Version, based on their sense of belonging, help-seeking style, and relationship with mental health. The participants were 608 Japanese public high school students. Of them, 129 (21.2%) and 479 (78.8%) were in the high autistic traits and control groups, respectively. A multiple regression analysis was performed using the General Health Questionnaire (GHQ) as the dependent variable for the groups. The analysis revealed no differences between the two groups regarding grade, gender, physical illness, insomnia, or mental health status; however, they differed regarding avoidant help-seeking style and teacher acceptance. Moreover, avoidant help-seeking scores in the high autistic traits group and teacher acceptance showed a significantly positive and significantly negative association with GHQ, respectively. The results indicated that children with autistic traits internally suppress them, experiencing distress. Teachers are aware that these students seek support from them, but the students seem reluctant to ask them for help. This can negatively impact the mental health of children with high autistic traits.

## 1. Introduction

Developmental disorders, such as Autism Spectrum Disorder (ASD) and Attention-deficit Hyperactivity Disorder, are most often diagnosed during early childhood and elementary school when the disorder’s characteristics are more apparent. Research indicates that early-age diagnosis accompanied by early rehabilitation preserves the quality of life of the individual and their family [1,2]. In Japan, it is crucial to verify the developmental potential for Autism Spectrum Disorder (ASD) at local health centers and medical institutions during the mandated 1–6 month and 3-year-old child health checkups. This process helps ascertain whether the child requires access to specialized medical care and education [3].

The diagnostic criteria for ASD were revised in the fifth edition of the *Diagnostic and Statistical Manual of Mental Disorders* (DSM-5) in 2013. The core symptoms of ASD include deficits in the use and understanding of social communication and interaction and restricted, repetitive patterns of behavior, interests, or activities [4]. Many studies have applied quantitative measures to ASD symptoms and characteristics and state that ASD characteristics are continuously distributed throughout the human population, forming a spectrum [5,6].

ASD has both mild and severe symptoms, and as the onset of symptoms varies throughout life [7], a certain number of people with ASD go through adolescence and young adulthood without a diagnosis of ASD, develop depression or adjustment disorder as a secondary disorder, and are subsequently diagnosed with ASD [8]. Symptoms of developmental disorders appear in their most basic form during childhood but are less noticeable after adolescence due to interactions with the environment and personal adaptation. Additionally, an ASD diagnosis requires the patient to show “signs of a disorder from childhood”. However, these signs may disappear when the patient reaches adulthood, or there may be a lack of information about the patient’s early years. In such cases, an adult diagnosis of ASD becomes challenging [9,10].

Certain individuals do not meet the clinical diagnostic criteria but exhibit strong social communication difficulties and sensory differences, which are among the characteristics of ASD. These individuals need psychosocial and educational support for obstacles in their school and daily lives, even if they are not receiving clinical treatment. Essentially, “undiagnosed” includes individuals who are “merely undiagnosed”, as well as those who do not meet the diagnostic criteria but have a strong sense of distress, referred to as “under the threshold” or Broader Autism Phenotype (BAP) [11]. BAP do not meet the diagnostic criteria for ASD in terms of social, cognitive, and personality characteristics but have been researched as a population with extensive ASD characteristics in terms of ASD diagnosis and tendencies within their blood relatives [12].

In Japan, the expression “gray zone” is occasionally used to refer to individuals who are “under the threshold” or BAP [13]. In 2021, the Ministry of Internal Affairs and Communications stated in its “Survey on University Support for Students with Disabilities, etc.—Focusing on Developmental Disabilities—” research report that some universities will consider applications for academic consideration from students who have not been diagnosed by a doctor, referred to as gray zone students. The justification provided emphasizes the need to support these students from an educational perspective, stating the following: “Based on the assumption of fluctuations in the boundary between disability and non-disability, support will be considered and necessary consideration will be granted based on the student’s individual sense of difficulty in studying, not based on a doctor’s diagnosis” [14].

Kamio et al. focused on the population “under the threshold”, or those who do not meet the diagnostic criteria but have as much social difficulty as individuals within the threshold [9]. The survey results of 22,529 Japanese children and adolescents indicate that behavioral characteristics are distributed on a single behavioral dimension—the autism spectrum [6]. The survey showed that regardless of whether they are on or below the diagnosis threshold, children and adolescents will have difficulty adjusting to society if they exhibit a certain level of autism spectrum characteristics. Moreover, they may fail to adapt unless they acquire the appropriate compensatory skills, and their quality of life in adulthood will be significantly reduced. She emphasized the need for support, stating that a decline in the quality of life is particularly noticeable in cases that have not received appropriate support during childhood and adolescence [9]. Murakami, a psychiatrist, recommends taking a broad view of individuals who are under the threshold, considering it “a developmental disability characteristic partially present since childhood but not noticeable and unnoticed” [15]. Particularly, among individuals with high-functioning ASD (HFASD) with average intelligence, the group that received an early diagnosis had a higher quality of life in adulthood than the group that received a delayed diagnosis [1]. This further underscores that even with undiagnosed ASD, delays in support, and treatment negatively impact prognosis.

The number of junior high and high school students in Japan who do not attend school has been increasing, and previous studies have indicated that individuals with ASD or strong ASD characteristics without a diagnosis contribute to this increase [16]. Some of the literature suggests that many people with HFASD or below the threshold do not have significant intellectual problems [17,18]. This is particularly true for those who pass the high school entrance exam and are admitted. During adolescence, friendships often hold greater significance than family relationships. This can present challenges for individuals with high autistic traits even below the diagnostic criteria, as they may encounter communication difficulties, struggle to keep pace with conversations among peers, and find themselves isolated due to challenges in navigating group dynamics [19]. It has also been noted that high school students become aware of their own characteristics, and their self-esteem declines. This is largely due to a negative self-image because of interpersonal differences and a sense of inadequacy over perceived ineffectuality [20]. According to a survey by Japan’s Ministry of Education, Culture, Sports, Science and Technology, “maladjustment at school entry and promotion” was the reason for non-attendance in 1.7% of elementary school students and 4.1% of junior high school students. However, the rate was highest at 9.4% for high school students [21]. This indicates that a key success factor is how well one adapts to school and finds a safe place; in other words, whether or not one can gain a sense of belonging. Although students with ASD have been found to have difficulty achieving a sense of belonging in school [22], there is no information regarding whether the situation is similar for students under the threshold.

Although it is useful to understand cognitive characteristics through psychological testing, individuals under the threshold are often overwhelmed by excessive testing because results can be inconclusive and inadequate for diagnosis. Murakami states that it is better not to be concerned with an exact diagnosis and that “rather, support should be started immediately as a ‘gray’ or ‘partial’ developmental disability” [15]. Murakami further stated that the prognosis of developmental disabilities depends not on the severity of the disability but on whether the patient has learned to receive help. He mentioned that for people with high autistic traits, the ability to have a help-seeking relationship with those around them will determine their trajectory in life; for individuals under the threshold with developmental disabilities, the ability to accept appropriate help-seeking actions is critical to their lives. Although children with ASD have difficulty engaging in appropriate help-seeking behaviors [23], there has been no research on what characteristics of help-seeking styles are present in children under the diagnostic threshold [9].

Based on his experience in supporting adults with undiagnosed ASD, Okamoto [24] indicated that challenges related to developmental disabilities do not suddenly manifest; rather, they become prominent cases when various challenges converge and surpass the individual’s capacity to cope. Therefore, it is critical to prevent health problems that may arise. He also stated that in preventive interventions, secondary disorders like anxiety, depression, withdrawal, and violence are often more significant issues than the core symptoms of developmental disorders. Okamoto stressed the significance of involvement throughout adolescence. In Japan, research on the daily life and educational support for individuals who are undiagnosed but strongly exhibit ASD characteristics—those who are referred to as below threshold or BAP, and in the “gray zone” in Japan—is extremely limited. Also, there is sparse research [25] on elementary school children as “children of concern” as well as university students [26]. In cases of undiagnosed ASD, social problems develop during adolescence and lead to secondary disorders. By the time students reach high school, a period when decisions regarding employment or higher education are critical, they need to be in a safe place where they can comprehend their own characteristics, employ help-seeking behaviors, and receive support. However, to our knowledge, there is no research targeting high school students who have strong traits for ASD, even though they have no intellectual problems and have not been diagnosed with ASD. This study explored the characteristics of school belonging and help-seeking styles of high school students who exhibit high autistic traits in comparison to other students. Moreover, the study examines their relationship with mental health to speculate on interventions that support students with high autistic traits, including preventing secondary disabilities.

## 2. Methods

### 2.1. Sample and Data Collection

In 2016, we conducted a cross-sectional, unmarked, self-administered questionnaire survey on 671 students in grades 1–3 (ages 15–18) attending a public high school in Tokyo, Japan. Student responses were collected from the school’s survey collection box, and 629 responses (collection rate = 93.7%) were received. Some of the data collected in this study were used in the author’s existing study [27], which has already been published in a journal. That study used variables not used in the present study as the main outcomes for approximately 120 pairs of matched parents and children. None of these participants (high school students) had been diagnosed with ASD or reported to the school that they required reasonable accommodation.

### 2.2. Ethical Considerations

This study complied with the 2023 amended guidelines of the Declaration of Helsinki. This study was approved by the Toho University School of Nursing Research Ethics Review Committee (Approval No. 28019; 27 October 2016). Since high school students participated in the study, per the ethical guidelines of the Japanese Ministry of Health, Labor, and Welfare, a written explanation was provided to each parent regarding their child’s participation. Parents were informed that high school student’s participation in the survey was voluntary and that there would be no disadvantages if they refused to participate. We assured the parents in writing that they could opt out of the study at any time through email, telephone, fax, letter, interview, or other means. Since this was originally a parent–child survey, students and parents were given a common ID. Parents were given opt-out forms from their children through the school. If parents did not consent to their child (student) cooperating with the survey, they could write their ID number on the enclosed document and mark a checkbox saying, “I do not consent”, which was returned to the researcher. They were also provided with a return envelope, although none of the parents refused to participate. ID numbers were assigned arbitrarily and did not correspond to student ID numbers or other numbers; this was also explained to the parents in writing. We made it clear to the students that their participation was voluntary and that they would not be disadvantaged if they did not participate. All measures were performed in accordance with the relevant rules, guidelines, and regulations.

### 2.3. Questionnaire Variables (Items)

Students were asked to note their gender and grade.

#### 2.3.1. Physical Problems

Participants were asked to indicate “what kind of physical problems they have had in the past month” from a list of 13 items, including headache, abdominal pain, diarrhea, nausea, dizziness, fever, malaise, shoulder pain or stiffness, eye strain, shortness of breath, stress, worries, and injury. A score of 1 point was given for each ‘yes’ response; hence, the scores ranged from 0 to 13 points. Higher scores indicated poorer physical condition.

#### 2.3.2. Athens Insomnia Scale

The Athens Insomnia Scale is a universal scale for determining insomnia created by the World Project on Sleep and Health, established by the WHO [28]. The scale assesses sleep quality and duration. Eight questions were quantified, with a maximum score of 24; higher scores indicate greater insomnia. The reliability and validity of the Japanese version of the scale have been verified [29]. Cronbach’s alpha coefficient was 0.85.

#### 2.3.3. Help-Seeking Behavior

The help-seeking behavior scale was developed by Nagai et al. It measures three help-seeking styles: independent help seeking (four items), excessive help seeking (four items), and avoidance of help seeking (four items). Its reliability and validity have been confirmed [30]. Scores ranged from 7 to 28 points for each style and are measured on a 7-point Likert scale, with higher scores indicating the aid request style. Assistance-seeking styles have been shown to be associated with depression. Cronbach’s alpha coefficient was 0.80.

#### 2.3.4. School Membership Scale

We used the shortened Japanese version of the School Membership Sensory Scale developed by American educational psychologist Goodnow [31] and colleagues. It consists of 13 items covering student acceptance (5 items), teacher acceptance (4 items), and sense of belonging to the school (4 items). Togari et al. confirmed the reliability and validity of the Japanese version of the scale [32]. The survey uses a 5-point Likert scale, with a higher score indicating a favorable situation. Cronbach’s alpha for this study was 0.816.

#### 2.3.5. Autism Spectrum Quotient Japanese Version (AQ-J-16; Abbreviated Version)

The Autism Spectrum Quotient (AQ) developed by Baron-Cohen et al. [33] is a 50-item self-administered questionnaire for normally intelligent adults designed to capture the autistic nature of the general population. It is intended to screen for high-functioning pervasive developmental disorders in people with an intelligence quotient of 70 or higher. The AQ-J-16 used in this study is a 16-item shortened version of the Japanese scale validated by Kurita, Koyama, and Nagata [34]. The items were rated on a 4-point Likert scale. Participants were given a score of 1 or 0 for each item (1 = high autism tendencies; 0 = low autism tendencies). A score of 12 points or higher indicates a strong autistic tendency but scoring 12 points or higher does not imply an ASD diagnosis, as this scale is not used for diagnostic implementation or screening. Seven items assessed poor communication skills; four, poor imagination; three, difficulty switching attention; and two, poor social skills.

#### 2.3.6. General Health Questionnaire-12 (GHQ12)

The GHQ is a questionnaire designed for the detection, symptomatology, assessment, and evaluation of clients with neurotic and depressive tendencies. It assesses nonspecific psychological distress and has been validated for its high reliability and construct validity [35]. In this study, we used the Japanese version with 12 items (hereafter referred to as GHQ-12), which was translated by Nakagawa et al. and has been verified for reliability and validity [36]. The GHQ-12 is effective as a screening tool for measuring mental health and has been validated, with no significant differences according to sex, age, education, or country. The questionnaire items were arranged on a 4-point Likert scale (0, I could do it; 1, I could not do it as well as usual; 2, I could not do it better than usual; 3, I could not do it at all); the more mentally unhealthy a person is, the higher the score. The total score was obtained using the Likert method in this study. The distribution of scores ranged from 12 to 48, with a Cronbach’s alpha coefficient of 0.820.

### 2.4. Statistical Analysis

The focus of this study was the total score of AQJ-16, which was confirmed for 614 students without any deficiency in AQJ-16. The mean score was 6.53, the median was 6.5, and the standard deviation was 2.47. Fifteen participants (2.5%) scored above the cutoff of 12, which is recommended by Kurita et al. [34]. None of the students in the targeted high schools were diagnosed with ASD, and there is a lack of research that shows where AQ-J scores classify people who are undiagnosed but have strong ASD characteristics. Therefore, a mean value of 6.53 plus a standard deviation of 2.47 was used to establish a group with a score of 9 points or higher in this study, which was analyzed as the “high autistic traits” group. In other words, the “high autistic traits group” in this study was defined as “a group of students with strong ASD characteristics but no diagnosis of ASD”. A total of 130 respondents (21.2%) scored nine or higher.

Based on this classification, we compared and analyzed the attributes, characteristics, and measurement scale scores between the high autistic traits group (AQ-J-16 score of 9 or higher) and the control group using the *t*-test for continuous variables and χ^2^ test for qualitative variables. Next, multiple regression analysis was performed for the high autistic traits group and the control group, with the GHQ as the dependent variable. 

There were 2 groups of 119 participants without missing values and 444 participants in the control group. Forced explanatory variables were gender (1 for males and 0 for females), total number of physical problems, Athens Sleep Scale score, and the three help-seeking styles—excessive, avoidant, and independent. The total score was used as input. For the School Membership Scale, each of the three subscales (Accepted by Students, Accepted by Teachers, Belonging to School) were submitted.

The only GHQ-related factor that differed between the two groups was the avoidant help-seeking style, which was significantly negatively associated only with the high autistic traits group. Considering the need to examine the interaction between strong ASD tendencies and aversive help-seeking styles, we performed a hierarchical multiple regression analysis with the GHQ as the dependent variable for everyone; the interaction term between these two variables was entered into the final model. To perform multiple regression analysis, including interaction terms for the AQJ-16 and Avoidant help-seeking style, centralization (subtracting the mean of each variable from each variable) was performed [37]; the respective variables and interaction terms were created and fed into the regression equation. In the hierarchical multiple regression analysis, the total AQ-J16 scores were entered as Model 1, in addition to the variables entered in the high autistic traits group and control groups. Model 2 introduced the interaction term between AQ-J16 and the Avoidant help-seeking style. The extent of change and significance of R2 was then assessed in Models 1 and 2, in addition to calculating R2.

To see what pattern of interaction was detected in the results of the hierarchical multiple regression analysis, we followed Aiken & West [38]. We substituted the “mean score ± standard deviation” for the focal variable—aversive help-seeking style. The adjustment variable (AQJ-16) in the regression equation obtained by the estimation equation values were substituted, and the calculated figures are shown in Figure 1.

## 3. Results

Table 1 compares attributes, characteristics, and measurement scale scores between the high autistic traits group and the control group. There were no differences in the proportions of males and females or grade levels between the two groups. The two groups had no statistically significant differences in the number of “Yes” responses regarding physical discomfort or the Athens Insomnia Scale scores.

Concerning help-seeking style scores, no significant differences were found between the two groups for excessive- and avoidant-style behaviors. However, the control group (AQ-J16 ≤ 8) had higher independent-style scores (18.49 ± 5.96, *p* = 0.024; the high autistic traits group score was 17.16 ± 6.02).

Regarding the school membership scale, there were significant differences between the two groups on the two subscales, except for acceptance by teachers. Acceptance by students (15.08 ± 3.18 and 13.83 ± 3.40, *p* < 0.001) and belonging to school (13.88 ± 2.84 and 12.40 ± 2.72, *p* < 0.001) showed significantly higher scores in the control group for the control group. No difference was observed for GHQ between the two groups.

The results of the multiple regression analysis with the GHQ as the dependent variable for each group are presented in Table 2. Since missing values were excluded, the high autistic traits group had 119 participants, and the control group had 444 participants for analysis. For the high autistic traits group, significant positive associations with GHQ (higher scores indicate poorer mental health) were found for physical problems (β = 0.188, *p* = 0.011), Athens Insomnia Scale (β = 0.288, *p* < 0.001), and avoidant help-seeking style (β = 0.236, *p* = 0.004). On the other hand, a significant negative association was found for Accepted by Students (β = −0.228, *p* = 0.001) and Accepted by Teachers (β = −0.175, *p* = 0.031). R^2^ was 0.468.

For the control group, significant associations with GHQ included physical problems (β = 0.146, *p* = 0.001) and the Athens Insomnia Scale (β = 0.324, *p* < 0.001), which were positive. The negative association was significant for Accepted by Students (β = −0.137, *p* = 0.001) and Accepted by Teachers (β = −0.088, *p* = 0.050); R^2^ was 0.388.

For the hierarchical multiple regression analysis to examine the interaction term, the GHQ was used as the dependent variable for all respondents. The results are summarized in Table 3. The AQ-J16 scores were included in Model 1, in addition to the variables shown in Table 2. In Model 2, the interaction term AQ-J-16 score × avoidant (help-seeking style) was entered. Significant positive associations with GHQ in Model 1 were physical problems (β = 0.151, *p* < 0.001), Athens Insomnia Scale (β = 0.314, *p* < 0.001), excessive help-seeking style (β = 0.090, *p* = 0.014), avoidant help-seeking style (β = 0.102, *p* = 0.006), and AQ-J16 (β = 0.080, *p* = 0.021). Significant negative associations were found between acceptance by students (β = −0.173, *p* < 0.001) and teachers (β = −0.118, *p* = 0.002). R^2^ was 0.411. Model 2 showed a significant positive association with the AQ-J-16 score × avoidant (help-seeking style) interaction term (β = 0.067, *p* = 0.039), although the significant variables did not change from those in Model 1. The explanatory power R^2^ for this model was 0.415, and the change in R2 was 0.004, which was significant (*p* = 0.039).

**Table 3 children-10-01927-t003:** Hierarchical multiple regression analysis with GHQ as dependent variable: an examination of the interaction between AQ-J and avoidant help-seeking style (n = 563).

Variables	Model 1					Model 2				
	Partial Regression Coefficient B	Coefficient β	*p*-Value	Standard Error	95%CI	Partial Regression Coefficient B	Coefficient β	*p*-Value	Standard Error	95%CI
Gender (male = 1, female = 0)	0.686	0.058	0.093	0.407	−0.114–1.487	0.698	0.059	0.086	0.407	0.100–1.497
Physical problems:number of items answered “yes”	0.353	0.151	<0.001	0.085	0.185–0.520	0.358	0.153	<0.001	0.085	0.190–0.525
Athens insomnia score (Higher scores indicate severe insomnia)	0.654	0.314	<0.001	0.078	0.501–0.806	0.646	0.310	<0.001	0.078	0.493–0.798
Help-seeking style										
Excessive (range 4–28)	0.081	0.090	0.014	0.033	0.016–0.145	0.080	0.089	0.0.15	0.033	0.0.16–0.144
Avoidance (range 4–28) *	0.090	0.102	0.006	0.033	0.026–0.154	0.089	0.100	0.006	0.032	0.025–0.153
Independent (range 4–28)	−0.003	−0.003	0.920	0.032	−0.066–0.060	−0.004	−0.004	0.903	0.032	−0.066–0.059
School Membership Scale										
Accepted by Students (range 4–20)	−0.313	−0.173	<0.001	0.082	−0.474–−0.150	−0.153	−0.171	<0.001	0.082	−0.469–−0.150
Accepted by Teachers (range 5–25)	−0.192	−0.118	0.002	0.062	−0.314–−0.070	−0.070	−0.117	0.002	0.062	−0.312–−0.070
Belonging to School (range 4–20)	−0.116	−0.057	0.189	0.088	−0.290–0.047	0.057	−0.062	0.151	0.088	−0.300–0.047
AQ-J16 (range 0.16) *	0.190	0.080	0.021	0.082	0.026–0.351	0.195	0.083	0.017	0.082	0.035–0.356
Added for Model 2										
AQ-J16×Avoidance {Help-seeking style)						0.025	0.067	0.039	0.0.12	0.001–0.050
R^2^		0.411					0.415			
						∆R^2^ = 0.004 (*p* = 0.039)				

* Variables entered as interaction terms.

**Figure 1 children-10-01927-f001:**
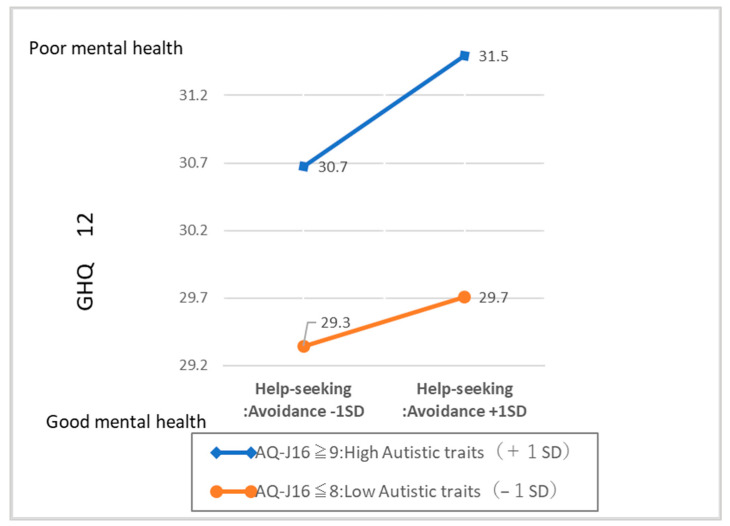
Interaction effects on GHQ in autistic traits and avoidance style of help seeking.

Figure 1 depicts the substituted values from the regression equation derived from the estimated equation. Higher scores on the aversive help-seeking style (a greater tendency to engage in aversive help-seeking behavior) were associated with higher GHQ scores (i.e., worse mental health) in the High Autistic Traits Group.

## 4. Discussion

No differences were found between the grey area and control groups in terms of grade, sex, physical illness, insomnia, or mental health status. Although the prevalence of ASD is significantly higher in boys [39], there was no difference between men and women when the ASD tendency was considered more broadly; this is a novel finding regarding the distribution by characteristics.

The lack of differences in physical discomfort, complaints of insomnia, and physical and mental illnesses are difficult to observe externally. Hence, they tend to be overlooked in school settings due to the apparent lack of observable characteristics. Originally, the hallmark of the autism spectrum was non-visibility [40], i.e., a deficiency in the brain. Even students with high ASD tendencies may not exhibit characteristic addictive behaviors or idiosyncratic remarks if the disorder is not severe enough to be diagnosed. They may appear comfortable and exhibit minimal discomfort or affect when teachers view the class as a whole. However, Table 1 shows that students with ASD tendencies did not feel accepted by their friends and felt out of place at school compared to the group without ASD tendencies.

Furthermore, the results showed that appropriate aid-seeking behavior for the ASD tendencies group was less effective than in the control group. This means that when they felt uncomfortable around their friends or in the classroom, they may have been unable to verbalize their feelings appropriately or communicate their problems to others. These results indicate that even if students with ASD feel out of place and do not fit in with their peers, this is not outwardly evident as it is an internal feeling. Consequently, it becomes exceedingly challenging for teachers to identify and offer appropriate support.

The hierarchical multiple regression analysis results indicated that for students with ASD tendencies, the relationship between and the presence of teachers was significant and closely related to mental health. Considering the school membership scale used in this study, items related to acceptance by teachers included “There is one teacher at school who listens to me when I am having trouble”, “Teachers are interested in me”, “I am treated as important as other students”, “Teachers care about me”, and a reversed item, “Teachers are not interested in me”. These findings demonstrate the importance of communicating with students, treating them fairly, and caring for them. Students with ASD tendencies often have difficulty finding an appropriate psychological distance from their peers [41,42], which may prevent them from establishing good relationships with their peers. On the other hand, teachers are mentally more mature adults than their peers and have better skills in caring and being considerate of others, so students with ASD characteristics may feel more comfortable in their relationships with teachers [43].

Moreover, high school students must consider their post-high school career paths. The surveyed schools exhibited a variety of options, including vocational schools, direct employment, and college. Their career choice is based on a comprehensive consideration of various factors, including the students’ wishes, academic ability and aptitude, personality traits, and financial situation—the more diverse the options, the greater the teacher’s role. Johnson [44] stated that teachers are expected to play a more supportive and guidance-oriented role for high school students with ASD tendencies. Choosing a career path that does not match their personality traits and aptitudes puts individuals with ASD at a higher risk of experiencing secondary disorders such as adjustment disorders and depression [45]. Teachers must observe students closely and interact with them during their daily class activities and tutoring sessions.

The adjusted R^2^ in the multiple regression analysis of the two groups showed a gap in scores between the two groups (0.468 and 0.388, both *p* < 0.001). The lower value in the control group compared to the other groups may be because mental health in high school students is generally associated with friendships, physical problems, academic stress, and family relationships [46], which may be strongly related. In the future, it is necessary to design a survey that considers these and other items (acceptance from teachers, help-seeking style) that exhibited differences in the group with ASD tendencies in this study.

The difference between the two groups was more pronounced in avoidant help-seeking styles. Although no significant difference was found in comparing mean scores between the groups, the group with ASD tendency showed a positive and significant association when the GHQ was used as the dependent variable. Moreover, the beta score in the ASD group was the second-largest association value among the variables entered after the Athens Insomnia Scale score. Based on these results, we examined the interaction between the AQ-j16 and avoidant help-seeking styles. Stronger ASD tendencies and a tendency to adopt an avoidant help-seeking style have been associated with poorer mental health. Poor communication and social skills are characteristics of ASD [47]. Their difficulty in effective communication and reluctance to seek advice or share their troubles with others, even when facing difficulties, suggests that they tend to keep their challenges to themselves. For example, it can be assumed that individuals with ASD tendencies will try to manage uncomfortable situations or troubling events in the classroom by avoiding consultation.

This idea that the cause of an individual’s problems is the individual themselves is called internalization. It has been shown that internalization causes individuals to view uncomfortable or unpleasant feelings as their own problems and interpret them as their own fault [48]. Some consider viewing these problems as one’s own responsibility as a coping mechanism because it “brings events within one’s control” and is therefore less altruistic. Alternatively, it can be interpreted as a strong sense of self-responsibility [49]. However, identifying the causes of what happens to oneself can contribute to self-criticism, low self-esteem, and increased feelings of isolation. If this mental state is not rectified, the result may be further instability.

Depression and adjustment disorders, which are considered secondary ASD disorders, have been found to have a significant impact on the lives of people living with ASD [8]. The prevention of secondary disorder onset is a notable issue. School teachers should be aware that students with ASD tendencies exist but that their challenges may not be immediately apparent. Moreover, teachers should note that students with ASD may struggle to seek help and try to handle their difficulties on their own, which can worsen their mental health. Systems should be introduced within the school to take corrective measures before the situation worsens.

### Limitations

This study was based on the results of only one high school in urban Japan; therefore, caution should be exercised when generalizing the results. The results may vary depending on whether the school is private, public, middle school and high school, or located in a rural area. Further research is required to confirm these results.

In addition, there is currently no clear definition of the concept of having strong ASD tendencies. In the current study, we incorporated the standard deviation (SD) into the analysis of AQ-J16, utilizing the mean to gauge ASD tendencies. It is important to note that the outcomes may vary when adjusting the range, either widening or narrowing it. This concept needs to be elaborated upon by experts and teachers in the future.

## 5. Conclusions

The results of this study show that the distribution of high school students who fall under the diagnostic threshold of ASD is unrelated to gender. Any complaints of physical illness and characteristics of poor physical condition have no notable difference to general students, but the difficulties they faced were confined to them internally. The concept of the “ASD gray zone” requires further discussion and research. A comfortable school for students with a diagnosis of ASD or high ASD tendencies is an environment where they can have a sense of belonging and that is appropriate for everyone. Teachers who engage with students should be mindful that some students exhibiting ASD tendencies seek a sense of belonging from their teachers. Additionally, some students may struggle to request assistance; within this group, those with ASD tendencies face greater challenges in speaking up. It follows that schools need to create a system whereby students can easily request assistance, and teachers can approach students carefully and consciously identify the needs they are struggling to convey.

## Figures and Tables

**Table 1 children-10-01927-t001:** Comparison of attributes and characteristics in groups with high and low autism spectrum tendencies.

Variables	High Autistic Traits Group AQ-J16 ≥ 9 *n* = 129	Control Group (Low Autistic Traits) AQ-J16 ≤ 8 *n* = 479	*t*-Test *p*-Value
	***n* (%), Average Score, ±SD**		
**Attributes and Characteristics**			
Male	66 (23.2)	219 (76.8)	0.273 (χ^2^-test)
Female	62 (19.3)	260 (80.7)	
1st Grade (15–16 years old)	46 (21.4)	169 (78.6)	0.761 (χ^2^-test)
2nd Grade (16–17 years old)	38 (19.6)	156 (80.4)	
3rd Grade (17–18 years old)	45 (22.6)	154 (77.4)	
Physical problems: number of items answered “yes”	2.74 ± 2.53	2.78 ± 2.46	0.690
Athens Insomnia Score	4.34 ± 2.90	4.22 ± 2.95	0.875
Help-seeking style			
Excessive (range 4–28)	10.26 ± 6.12	11.36 ± 6.67	0.091
Avoidance (range 4–28)	13.81 ± 6.66	11.36 ± 6.68	0.051
Independent (range 4–28)	17.16 ± 6.02	18.49 ± 5.96	0.024
School Membership Scale			
Accepted by Students (range 4–20)	13.83 ± 3.40	15.08 ± 3.18	<0.001
Accepted by Teachers (range 5–25)	14.91 ± 3.82	15.30 ± 3.59	0.290
Belonging to School (range 4–20)	12.40 ± 2.72	13.88 ± 2.84	<0.001
GHQ (range 12–48)	27.17 ± 6.07	26.26 ± 5.78	0.132

Where the number of people did not match the total, it was a missing value.

**Table 2 children-10-01927-t002:** GHQ-related factors by two groups.

	High Autistic Traits Group n = 119						Control Group (Low Autistic Traits) n = 444)					
	AQ-J16 ≥ 9						AQ-J16 ≤ 8,					
Variables	Univariate		Multivariate				Univariate		Multivariate			
	Coefficient β	*p*-Value	Coefficient β	*p*-Value	Standard Error	95%CI	Coefficient β	*p*-Value	Coefficient β	*p*-Value	Standard Error	95%CI
Gender (male = 1, female = 0)	0.120	<0.001	0.004	0.951	0.877	−1.686–1.793	0.182	<0.001	0.069	0.084	0.466	−0.108–1.722
Physical problems: number of items answered “yes”	0.319	<0.001	0.188	0.011	0.173	0.103–0.788	0.407	<0.001	0.146	0.001	0.101	0.138–0.536
Athens insomnia score (Higher scores indicate severe insomnia)	0.491	<0.001	0.288	<0.001	0.152	0.292–0.893	0.497	<0.001	0.324	<0.001	0.092	0.498–0.857
Help-seeking style												
Excessive (range 4–28)	0.002	0.979	0.130	0.100	0.076	−0.025–0.278	0.065	0.155	0.077	0.068	0.037	−0.005–0.140
Avoidance (range 4–28)	0.450	<0.001	0.236	0.004	0.076	0.071–0.371	0.177	<0.001	0.066	0.120	0.037	−0.0.15–0.129
Independent (range 4–28)	0.109	0.223	0.066	0.336	0.068	−0.069–0.201	−0.118	0.0.10	−0.036	0.341	0.037	−0.107–0.037
School Membership Scale												
Accepted by Students (range 4–20)	−0.498	<0.001	−0.228	0.009	0.160	−0.742–−0.110	−0.429	<0.00.1	−0.173	<0.001	0.095	−0.501–−0.128
Accepted by Teachers (range 5–25)	−0.461	<0.001	−0.175	0.031	0.129	−0.536–−0.030	−0.322	<0.001	−0.088	0.050	0.072	−0.284–0.00.0
Belonging to School (range 4–20)	−0.384	<0.001	−0.026	0.757	0.190	−0.434–0.317	−0.40.1	<0.001	−0.079	0.115	0.102	−0.362–0.040
R^2^			0.468						0.388			
			*p* < 0.001						*p* < 0.001			

Where the number of people did not match the total, it was a missing value.

## Data Availability

The data presented in this study may be available on request from the corresponding author under certain conditions. The data are not publicly available due to the data from this survey, which included parents and children attending high schools in Tokyo, are highly sensitive, containing information about autism as it relates to legal minors. The surveyed schools are not allowed to disclose raw data, even anonymously.

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
