# Peer review of "A Sense of Belonging and Help Seeking: Examining Factors Related to the Mental Health of High School Students with High Autistic Traits without Diagnosis"

_children, 2023, doi:10.3390/children10121927_

Round 1

Reviewer 1 Report

Comments and Suggestions for Authors

Dear authors:

Thank you for allowing me to read and review your work. The problem of late diagnosis of autism is very interesting and the results you point out about the relationship between people who occupy the so-called "grey zone" supports the idea and opens to find explanations about the differences in diagnosis according to gender. Although this study is based on a single centre, I think it opens the door to replications with more representative samples.

I believe that the article, even with the limitations stated in it.

Best regards

Author Response

We thank you for your encouragement.

The expression "gray zone" is widely debated. Hence, we have reviewed the manuscript and revised it to make the discussion more precise, although the gist of the argument has not changed significantly.

As you pointed out, this study focused on a single institution, and there are problems with generalizability. As mentioned in the limitations of the study, I believe a larger study is needed.

Thank you for your valuable feedback.  

Reviewer 2 Report

Comments and Suggestions for Authors

Dear authors,

I appreciate you research, since it deals with an original topic, i.e. the sense of belonging and help-seeking: examining factors related to mental health in high school students in the Autism Spectrum Disorder “gray zone”.

The manuscript is clear, relevant for the field and presented in a well-structured manner.

The manuscript is scientifically sound, and the experimental design is appropriate to test the hypothesis. Statistical analysis employed are well explained.

The data is interpreted appropriately and consistently throughout the manuscript.

The conclusions are consistent with the evidence and arguments presented.

I would like to suggest you only some minor revision in "Introduction" paragraph.

You could briefly describe the core symptoms of ASD when you introduce it, according to DSM-5 (DSM-5-tr).

Moreover, I would suggest you some papers to deepen the importance of early diagnosis in ASD. You mentioned it, but I think this topic should have more relevance in your paper. Early diagnosis of ASD is of crucial importance, allowing effective intervention strategies aimed at symptom management to be implemented at an early stage. In this case, even those who would present a risk profile could benefit from intervention measures aimed at positively impacting their development. 

You can find some information about this topic in these papers:

Boccaccio, F. M.; Platania, G. A.; Guerrera, C. S.; Varrasi, S.; Privitera, C. R.; Caponnetto, P.; Pirrone, C.; Castellano, S. Autism Spectrum Disorder: Recommended Psychodiagnostic Tools for Early Diagnosis. Health Psychol Res 2023, 11, 77357. https://doi.org/10.52965/001c.77357.

Elder, J. H.; Kreider, C. M.; Brasher, S. N.; Ansell, M. Clinical Impact of Early Diagnosis of Autism on the Prognosis and Parent–Child Relationships. Psychol Res Behav Manag 2017, 10, 283–292. https://doi.org/10.2147/PRBM.S117499.

Author Response

 We thank you for reviewing our research paper and providing us with your opinions.

The expression "gray zone" is highly debated. Hence, we have reviewed the manuscript and revised it to make the discussion more precise, although the gist of the argument has not changed significantly.

We thank you for your suggestions for additions to the introduction. I believe that the references you provided complement the basis of the paper, and have cited them as references 2 and 7.

Reviewer 3 Report

Comments and Suggestions for Authors

Line 43 – 44 - spectrum of 43 “normal” and “abnormal” is old and considered inappropriate for a scientific paper and language

Line 44 – 47 – I do not think the example cited proves the idea of having “normal” and “abnormal” in the population

Line 48 – 52 – I find this statement inappropriate – this is not the idea of atypical autism; the authors can check the classification systems for better understanding of what atypical autism is and they can try to understand the transition to the idea of “spectrum”. Also, I do not understand what “typical autism” is. There is no such a category.

Line 53 – 54 – I disagree with the idea of gray zone diagnosis in general, even if the authors can cite some other papers, this term is offensive to some people, and scientifically unclear, especially considering that ASD is a diagnosis that is present in classifications, the criteria are clear, and also the process by which a diagnosis of a mental disorder is made from a clinical point of view point is too clear.

Line 55- 56 – This is not true

Line 57 – 60 – This is not true

Line 61 – 64 – Even if there is a definition cited, this is not true and doesn’t meet the clinical categories. If some people have some symptoms, then there are 2 opportunities – 1 – there is no diagnosis of mental illness /I do not think “gray zone” can be acceptable idea, since all of us will participate in 1 or more “gray zones” of 1 or more mental diagnoses/ and 2 – there is another disorders /the authors can check the publications about DD of ASD and other mental diseases – this is clear/.

Line 61 – 80 - It looks like we are going to hunt people down and look for what was undiagnosed in their childhood in order to diagnose them with a mental disorder as adults.

Linen 83 – 85 – This is not true

Line 97 – 98 – I find this a wrong interpretation

Line 106 - ‘gray’ or ‘partial’ developmental disability.” Sounds unscientific, clinically wrong, offensive, harmful to science and practice

Line 119 – 121 – those could be primary, not secondary to the no existing “gray zone” of ASD

The Autism Spectrum Quotient (AQ) is not a diagnostic instrument, neither point the idea of “gray zone” autism

Line 226 – 228 – “the study added” sounds quite worrying.   By using an instrument that measures something entirely different, the authors add a measurement that proves their unconfirmed claim of an available category of mental disorder

Table 1 and below – Include non-existing categories

Since the study contains some interesting facts, I would strongly recommend that the authors transform it in general, remove the gray areas like ideas, leave only what the AQ measures and make comparisons with the other categories they include. A diagnosis of ASD can be established by science- and practice-clear methods that are widely cited. Of course, this means a complete reworking of the first part of the article, but in this form, I think it definitely does not meet the scientific and ethical requirements for publication.

Author Response

Thank you for your comments.

We have considered your points very seriously and have held extensive discussions with a new researcher who specializes in supporting people with developmental disabilities in educational institutions.

First, we reviewed the DSM-5 and included its definitions in the revised draft to clarify the topic of the study.

Additionally, we checked for the expressions "gray zone" and "gray area" in the Japanese medical literature database and other databases related to international medical papers. We believe that although the expression "gray zone" is commonly used among Japanese researchers and teachers in health and education as an ambiguous expression or idiom, it is not a medical term and may be inappropriate as the central concept for paper published in an international journal.

Although the term "gray zone" is occasionally used in the revised manuscript, we clarify that it is "occasionally referred to as such in Japan".

After discussing the above, the following modifications were made, mainly to the introduction.

The study focuses on students who score 9 or higher on the AQ-J16, who are below the diagnostic threshold but may have high autistic traits.

These students are not diagnosed and probably do not meet clinical criteria either. However, these students may need assistance in school life and academics because of their high autistic tendencies, even if they do not meet the diagnostic criteria. (Kamio et al., 2013; Kanto Government District Administrative Evaluation Bureau. Ministry of Internal Affairs and Communication, 2020; Murakami, 2023)

Some of the authors have been involved in the care of "children who have not been diagnosed but have strong ASD characteristics" for many years in public middle and high schools. Usually, teachers believe that they do not need to provide support and attention to a student without a diagnosis. However, some students did have difficulties due to their strong ASD characteristics and needed considerable support in the absence of a diagnosis. (Kamio et al., 2013; Kanto Government District Administrative Evaluation Bureau. Ministry of Internal Affairs and Communication, 2020; Murakami, 2023). 

We would greatly appreciate it if you could confirm the above, and clarify if our revisions have adequately addressed your concerns. Your input has helped us better understand our research and has clarified ambiguities and shortcomings in this study, making it more meaningful. Thank you very much.

<References ※Mainly Written in Japanese >

Kamio, Y.; Moriwaki, A.; Takei, R.; Inada, N.; Inokuchi, E.; Takahashi, H.; Nakahachi, T. Psychiatric issues of children and adults with autism spectrum Psychiatric issues of children and adults with autism spectrum disorders who remain undiagnosed. Psychiatr Neurol Jpn 2013: 60.1-06, 115

Kanto Government District Administrative Evaluation Bureau. Ministry of Internal Affairs and Communications. Survey Report on University Support for Students With Disabilities. - focusing on developmental disabilities-; Vol. 331, 2020.

Murakami, S. ‘Characteristics of adult developmental disabilities and “gray zone’ and points of support [Japanese]. The Jpn J Public Health Nurse 2023, 12–17.

Round 2

Reviewer 3 Report

Comments and Suggestions for Authors

Line 15 and 22 - "High Autistic Traits” group – What is High autistic traits? You must provide definition when you first use the term and define if this is a term that you want to present for a first time. This term is the core characteristic of your paper and even if it is not widely used, I agree that you have an opportunity to suggest it. Then you must describe it carefully when you mention it first in the text and be ready to defend the idea of its existence.

Line 98 – 99 “Some individuals with HFASD or who are under the threshold do not have significant intellectual problems” – Do you think some HFASD have significant ID?

Line 138 – I do not understand that statement “individuals who are undiagnosed but strongly exhibit ASD characteristics”

Line 172-173 – I do not understand that  “students completed a parent questionnaire, along with a written explanation, including a …”

Line 220 – 221 – You must emphasize that the questionnaire is not used for diagnosis. “but is not a diagnostic of ASD” – seems that this phrase is relevant to the number of points not to the instrument

Author Response

Thank you for your very useful suggestions for our study. We made revisions considering your suggestions. We also made many minor revisions in grammar and style without including a specific response.

We hope that the revisions in the manuscript and our accompanying responses will be sufficient to make our manuscript suitable for publication in Children.

The following are responses to each of the comments.

We look forward to hearing from you at your earliest convenience.

Reviewer number: 3

Reviewer's report:

Line 15 and 22 - "High Autistic Traits” group – What is High autistic traits? You must provide definition when you first use the term and define if this is a term that you want to present for a first time. This term is the core characteristic of your paper and even if it is not widely used, I agree that you have an opportunity to suggest it. Then you must describe it carefully when you mention it first in the text and be ready to defend the idea of its existence.

 Response: Thank you for this suggestion. I agree with your point that it is necessary to state the definition when the term is first used. Thank you for pointing it out. I have added a definition to the abstract. This term is also defined in the methods section. Again, thank you for your feedback.

Line 98 – 99 “Some individuals with HFASD or who are under the threshold do not have significant intellectual problems” – Do you think some HFASD have significant ID?

 Response: Thank you for your comment. I checked the literature again regarding this. Some literature states that HFASD does not have a significant intellectual disability, so we decided that it was necessary to cite the literature. Also, since the description was too conclusive, we decided that additional revisions were necessary. We would appreciate it if you could review them.

Cremone, I.M.; Carpita, B.; Nardi, D.; Casagrande, R.; Stagnari, G. Amatori and L. Dell'Osso. Measuring social camouflaging in individuals with high functioning autism: A literature review. Brain Sci 2023, 13. DOI:10.3390/brainsci13030469.

Dell'Osso, L.; Luche, R.D.; Gesi, C.; Moroni, I.; Carmassi, C.; Maj, M. From asperger's autistischen psychopathen to dsm-5 autism spectrum disorder and beyond: A subthreshold autism spectrum model. Clin Pract Epidemiol Ment Health 2016, 12, 120-31. DOI:10.2174/1745017901612010120.

 Line 138 – I do not understand the statement “individuals who are undiagnosed but strongly exhibit ASD characteristics”

 Response: I think the description was difficult to understand, thank you for pointing it out. Individuals who are undiagnosed but strongly exhibit ASD characteristics refers to someone below the threshold, or BAP, or what we call gray zone in Japan, so I added this information for clarification.

Line 172-173 – I do not understand that  “students completed a parent questionnaire, along with a written explanation, including a …”

 Response: Regarding this point, the description was very difficult to understand and could cause confusion. We added information and revised the text to ensure that the content was clearly conveyed, and checked the language with an English proofreading company. Thank you for pointing this out.

Line 220 – 221 – You must emphasize that the questionnaire is not used for diagnosis. “but is not a diagnostic of ASD” – seems that this phrase is relevant to the number of points not to the instrument

Response: Thank you for highlighting this. Indeed, as you pointed out, the description was very vague and difficult to understand. We have revised the information to clarify that it cannot be used for screening or diagnosis. Thank you for paying attention to these details.